# Designing Skill-Compatible AI: Methodologies and Frameworks in Chess

**Karim Hamade**
kar@cs.toronto.edu
University of Toronto

**Reid McIlroy-Young**
reidmcy@cs.toronto.edu
University of Toronto

**Siddhartha Sen**
sidsen@microsoft.com
Microsoft Research

**Jon Kleinberg**
kleinberg@cornell.edu
Cornell University

**Ashton Anderson**
ashton@cs.toronto.edu
University of Toronto

## Abstract

Powerful artificial intelligence systems are often used in settings where they must interact with agents that are computationally much weaker, for example when they work alongside humans or operate in complex environments where some tasks are handled by algorithms, heuristics, or other entities of varying computational power. For AI agents to successfully interact in these settings, however, achieving superhuman performance alone is not sufficient; they also need to account for suboptimal actions or idiosyncratic style from their less-skilled counterparts. We propose a formal evaluation framework for assessing the compatibility of near-optimal AI with interaction partners who may have much lower levels of skill; we use popular collaborative chess variants as model systems to study and develop AI agents that can successfully interact with lower-skill entities. Traditional chess engines designed to output near-optimal moves prove to be inadequate partners when paired with engines of various lower skill levels in this domain, as they are not designed to consider the presence of other agents. We contribute three methodologies to explicitly create skill-compatible AI agents in complex decision-making settings, and two chess game frameworks designed to foster collaboration between powerful AI agents and less-skilled partners. On these frameworks, our agents outperform state-of-the-art chess AI (based on AlphaZero) despite being weaker in conventional chess, demonstrating that skill-compatibility is a tangible trait that is qualitatively and measurably distinct from raw performance. Our evaluations further explore and clarify the mechanisms by which our agents achieve skill-compatibility.

## 1 Introduction

As AI achieves superhuman performance in an increasing number of areas, a recurring theme is that its behavior can be incomprehensible to agents of lower skill levels, or incompatible with their behavior. Game-playing is a familiar instance of this; it is well-understood, for example, that modern chess engines play in a style that is often alien to even the best human players, to the extent that calling out "engine moves" (actions only computers would take) is a staple of professional commentary. It is also standard practice for human players to use moves outputted by chess engines as recommendations for their own play, only to find that they cannot successfully follow them up. In a way, chess AI (which is "near-optimal") and human players (as examples of less-skilled agents) "speak" very different dialects that are often not mutually intelligible. This lack of compatibility leads to failures when less skilled agents interact with optimal ones, in settings where the less-skilled parties may be humans interacting with powerful AI systems, or simple heuristics interacting with much stronger ones.

Given this state of affairs, an important open question in any given domain is how to achieve AI performance that is *both* high-level and compatible with agents of lower skill. How might we accomplish this, and how would we know when we've succeeded?

In this work, we propose a training paradigm to create AI agents that combine strong performance with skill-compatibility, we use our paradigm to train several skill-compatible agents, and we illustrate their effectiveness on two novel chess tasks. Our paradigm is based on the following idea: skill-compatible agents should still achieve a very high level of performance, but in such a way that if they are interrupted at any point in time and replaced with a much weaker agent, the weaker agent should be able to take over from the current state and still perform well. We enforce a high level of performance by testing in an adversarial environment where the opponent might have superhuman abilities, and we encourage robustness against interruption and replacement by a weaker agent in chess (*skill-compatibility*) in two different ways: independently at random after each move; and by decomposing individual moves into a selection of a piece type and a subsequent selection of a move using that piece type, following the popular chess variant known as "hand and brain."

This interruption framework thus provides computationally powerful agents with an objective function that balances two distinct goals: (i) it dissuades them from playing incompatible "engine moves," since such moves may strand a weaker agent in a state where it can't find a good next action even when one exists, but (ii) it still promotes high performance since the goal is to win games despite interruptions from weaker counterparts. Our paradigm thus suggests a strong and measurable notion of the interpretability of a powerful agent's actions: the actions are interpretable (and skill-compatible) in our sense if and only if a weaker agent can find an effective way to follow up on them. This grounding is particularly useful in complex settings such as our motivating domain of chess which contains actions where there might be no succinct "explanation" (we know this both in practice, and also theoretically from the fact that general game-tree evaluation is believed to be outside the complexity class NP.)

## 2 Background and Related Work

**Chess and AI.** Chess has a long history in AI research Shannon (1950), with milestones including Deep Blue Campbell (1999), followed by superhuman performance on commodity hardware, and more recently AlphaZero and its follow-ups Silver *et al.* (2016); Schrittwieser *et al.* (2020). More recent work on the relation of algorithmically-generated chess moves to human behavior play an important role in our work McGrath *et al.* (2022); McIlroy-Young *et al.* (2020); Anderson *et al.* (2017); Maharaj *et al.* (2022).

**Human-AI collaboration.** Recent work has studied human-AI collaboration in a multi-agent scenario, where an AI agent and a weaker human agent work alongside each other to complete a task Carroll *et al.* (2019); Strouse *et al.* (2021); Yang *et al.* (2022). One distinction from our work is their notion of compatibility, where the focus has been on agents working simultaneously on related tasks; in contrast, a central feature of our framework is that the compatibility is *inter-temporal*, with the design goal that a less-skilled agent should be able to take over at any point from the partially completed state of the AI agent's progress.

**Opponent modeling.** Agents that interact with humans have made great strides in performance by modeling other human actors Bard *et al.* (2019), whether by modeling opponents as an optimal player (Perolat *et al.*, 2022; Gray *et al.*, 2020; Brown & Sandholm, 2018), or by building agents that communicate and collaborate with other human players in multiplayer games (Vinyals *et al.*, 2019; Yu *et al.*, 2021; (FAIR)† *et al.*, 2022). Several prior works explore from a strict adversarial, non-collaborative perspective the use of MCTS to exploit suboptimal play safely, in strong agents (Wang *et al.*, 2022; Ganzfried & Sandholm, 2015).

### 2.1 Chess engines

We select chess as our model system due to the ready availability of both superhuman AI agents and AI agents designed to emulate lower-skilled human players, the complexity of the decision-making task, and the need for more understandable game AI agents that others can interact with. We list the existing engines that we make use of in our work.

**leela.** leela Lyashuk & et al (2023) is an open source version of AlphaZero Silver *et al.* (2018), a deep RL agent that consists of a neural network that evaluates boards (value) and suggests moves (policy) Silver *et al.* (2016), both of which are used to guide a Monte Carlo Tree Search

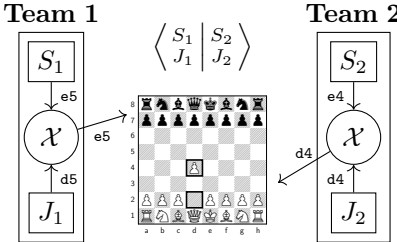
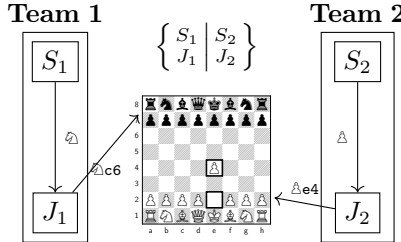

Figure 1: Stochastic Tag Team Framework        Figure 2: Hand and Brain Framework

(MCTS) algorithmto select the next action Jacob *et al.* (2022); Grill *et al.* (2020). The network is trained using repeated iterations of self-play followed by back-propagation. We use a small version of LEELA as the superhuman engine in our tests and as our baseline for comparison.

**MAIA.** The MAIA engines McIlroy-Young *et al.* (2020; 2022) are a set of human-like chess engines that capture human style at targeted skill levels. They are trained as a classification task on tens of millions of human games at a specified skill level, and as such MAIA does not use MCTS during play. Most of our work uses the weakest version, MAIA 1100, which was trained on games from 1100-rated players—those roughly in the 20th percentile of skill—on the open-source online chess platform lichess.org. MAIA serves as the instantiation of sub-optimal, lower-skilled agents.

## 3   METHODOLOGY

### 3.1   CHESS FRAMEWORKS

Our goal is to develop chess agents that can behave in inherently skill-compatible ways, much like skilled coaches tailor their actions to suit their students. In this work, we approach this task via two proxy frameworks in which our engine "coaches", or *seniors*, interact collaboratively with weaker "students", or *juniors*. There are two natural ways in which this collaboration could take place within a sequential game setup: the seniors and juniors could alternate between who takes the current action, or the seniors and juniors could collaboratively construct each action. Our two frameworks follow these two directions to enable cooperation within a chess game, suiting our experimental purpose of designing and evaluating skill-compatible agents. The former can be likened to self-driving cars, where AI needs to be ready for a hand-off to humans at a moment's notice, whereas the latter resembles automatic stock picking strategies where potential candidates are filtered by AI for humans to make final picks. Note that both frameworks incorporate a strong and weak engine jointly in control of a single color, and incorporate an element of stochasticity to preclude perfect prediction strategies by agents that skirt the objective of achieving compatibility. We will generally refer to our agents as playing the *Focal* roles against a team of opponent agents playing the *Alter* roles.

**Stochastic Tag Team ($STT$).** The $STT$ setup consists of two teams, each in control of a color on the chessboard. A team consists of two agents, the junior teammate, which is generally MAIA, and the senior teammate, which is generally a stronger engine (e.g. LEELA, or the engines we design). Prior to each move, Nature flips a fair coin to determine whether the junior or senior makes the move, with no consultation with any other party allowed. This setting introduces a cooperative aspect to chess, since a senior will need to be prepared for the possibility that a weaker junior might be making the next move. At the same time, the senior will need to play at a high level of chess, since the opponent senior will also be playing at a high level, and it will simultaneously need to attempt to exploit the weaknesses in the opponent junior. The $STT$ framework thus allows us to explore scenarios where teammates and opponents can be both high-skilled and low-skilled, and the high-skilled AI is required to both perform at a high level and account for low-skill involvement in both teams. See Figure 1. We use the following tuple to denote a game played under $STT$: $\left\langle \frac{S_1}{J_1} \middle| \frac{S_2}{J_2} \right\rangle$, where $S_1$ is the senior engine on the white team, $J_1$ the junior engine on the white team, $S_2$ the senior engine on the black team, and $J_2$ the junior engine on the black team. Note that while the first

team is technically white and the second team black in this notation, we abuse the notation to indicate multiple games played between these two teams, alternating between black and white.

**Hand and Brain ($HB$).** This cooperative setup, which has witnessed a massive increase in interest by grandmasters (GMs) and amateurs alike in recent years, also consists of two teams, each in control of a color on the chessboard. A team consists of two agents, the brain agent (always the stronger agent in our case), which selects the piece type to be moved (e.g., knight ♘), and the hand (MAIA agent in our case), which then selects the specific piece and move to make given the brain's selection (e.g., move the knight on g1 to f3). See Figure 2. GM Hikaru Nakamura stated that as the brain playing with a weaker hand, he often picks sub-optimal moves he finds more suitable for his hand partner Nakamura (2021), which is in the spirit of our work. In contrast to the previous framework, this framework exemplifies scenarios where the stronger agent "nudges" the weaker agent, narrowing their action space, which then makes the action. We will use the following tuple to denote a game played under $HB$: $\left\{ \begin{smallmatrix} H_1 \\ B_1 \end{smallmatrix} \middle| \begin{smallmatrix} H_2 \\ B_2 \end{smallmatrix} \right\}$, where $H_1$ and $B_1$ are the hand and brain agents on the white team, and $H_2$ and $B_2$ are the hand and brain agents on the black team. In our work, the brain will always be the stronger agent, and the hand the weaker engine that chooses a move (stochastically from its distribution, to induce some randomness and ensure the brains aren't able to perfectly predict their decision) conditioned on the brain's piece-type choice.

## 3.2 Methodologies to create skill-compatible seniors

We contribute three methodologies to create agents that outperform superhuman chess engines (LEELA) in these two skill-compatibility frameworks. We emphasize that we are not aiming to improve upon state-of-the-art chess engines. Instead, we are interested in designing skill-compatible chess AI that can productively interact with weaker agents.

**Tree agent.** Our first agent is the MAIA engine augmented with MCTS. By exploring future game states based solely on MAIA's policies and values, the TREE agent inherently takes its junior's propensities into account when deciding what move to make. Notably, this agent only requires a MAIA model and does not rely on a superhuman agent. It is also framework-agnostic and thus implemented identically for both frameworks. In $HB$, the TREE agent's output is filtered to only convey the piece type, as mentioned above.

**Expector agent.** We introduce a type of gold standard agent that conforms to the exact setting of the frameworks. The expector agent has access to models of juniors/hands, and maximizes its expected win probability $w$ over a short time horizon given the identities of the seniors and juniors. For $STT$, it simulates all possible bitstrings 2 plies into the future, selecting the move $m = \underset{m}{\text{argmax}} \left( \mathbb{E}_{s \in \{00,01,10,11\}} [w | (m,s)] \right)$; and for $HB$ it selects the piece that maximizes its expected win probability over MAIA's distribution of moves conditioned on that piece ($D_p$): $p = \underset{p}{\text{argmax}} \left( \mathbb{E}_{m \in D_p} [w | m] \right)$. In the former case it requires a model of the other three agents in the game to perform the simulation, and in the latter case it also needs a model of its own hand to obtain the distribution $D_p$. In both cases, it requires access to a strong agent to compute the win probabilities that the expectation is maximizing. Although it requires no training, playing moves is expensive due to calls to multiple chess engines and evaluations of the current board state. The version of the expector designed for $STT$ will be denoted as $\text{EXP}_\text{T}$, and the one designed for $HB$ will be denoted as $\text{EXP}_\text{H}$.

**Attuned agent.** The attuned agent is a self-play RL agent that directly learns from playing in the two frameworks. In contrast to learning from self-play in conventional chess, as LEELA does, the attuned agents are created by generating games from self-play of LEELA and MAIA teams in both frameworks. With a small training set, this method is essentially a fine-tuning procedure for LEELA that takes into account MAIA's interventions, rather than training a chess engine from scratch. The Supplement includes full training details. This methodology is the most practical of the three in its ability to be modified and generalize, and lies somewhere between the rigidity of TREE and the specificity of EXP. However, it is the only agent of the three we introduced that requires training. The version of the attuned agent trained on $STT$ will be denoted as $\text{ATT}_\text{T}$, and the one trained on $HB$ will be denoted as $\text{ATT}_\text{H}$. In $HB$, similar to the tree agent, we only take the piece of the outputted move.

Table 1: Game Results for all agents $STT$ and $HB$

| Game Setup | | $STT$ | | | | $HB$ | | | |
| --- | --- | --- | --- | --- | --- | --- | --- | --- | --- |
| | | TREE | EXP$_\text{T}$ | ATT$_\text{T}$ | MAIA | TREE | EXP$_\text{H}$ | ATT$_\text{H}$ | MAIA |
| focal MAIA | LEELA MAIA | 56.5$^{**}$ | **66.5$^{**}$** | 55.0$^{**}$ | 7.5$^{*}$ | **60.0$^{**}$** | 56.5$^{**}$ | 55.0$^{**}$ | 27.5$^{*}$ |
| focal | LEELA | 0.5$^{**}$ | 14.0$^{*}$ | 12.5$^{*}$ | 0.0$^{**}$ | 0.5$^{**}$ | N/A | 4.5$^{*}$ | 0.0$^{**}$ |

$^{**} \leq \pm 0.5$, $^{*} \leq \pm 1.5$, see appendix for full error ranges. Note that EXP$_\text{H}$ doesn't output moves, so can't play leela directly.

## 4 EXPERIMENTS

### 4.1 AGENT STRENGTH IN EACH FRAMEWORK

Our foremost goal is to quantify the objective performance of each agent on each framework. Are they better at playing with weaker partners than state-of-the-art chess AI—that is, are they skill-compatible? Here, we use MAIA1100 as the junior and hand agent for all analyses, and the three methodologies will accordingly make use of MAIA1100 as a base model to guide the search for TREE, as a junior to guide the look-ahead for EXP, and in the creation of the training dataset for ATT. Our evaluation metric is the win-share over games, which is defined as $\frac{(W+D/2)}{n}$, where $W$ is the number of wins, $D$ the number of draws, and $1000 \leq n \leq 10000$ the number of games, from the perspective of the focal team. Equally-matched agents will each score a win-share of 50%, and scoring above 50% indicates a victory. We compute the standard error by treating the experiments as random samples from a trinomial distribution, as detailed in the Supplement. Tables 1 shows the results for all three agents on both frameworks $STT$ and $HB$ respectively.

As shown in the first row, which documents the performance of our focal agents in matchups against the state-of-the-art chess AI LEELA in our frameworks, all of our methodologies achieve a winning score ($>50\%$) in both frameworks. In $STT$, the EXP$_\text{T}$ agents that perform a short look-ahead (66%) tend to dominate, and in $HB$ it is the TREE agent that scores the highest (60%). Our main result is that all three methodologies produce agents that play well *and* more intelligibly to weaker partners than state-of-the-art chess AI.

In order to validate that our focal agents are achieving their gains by explicitly accounting for the presence of the weaker partners, we eliminate the two most pressing potential confounders. The first hypothesis we rule out is that our focal agents are simply stronger than LEELA, which the second row in Tables 1 shows is evidently not the case. In fact, they are significantly weaker, losing most of their games to LEELA in head-to-head regular-chess matchups. TREE performs particularly poorly ($<1\%$), likely because it is unrelated to LEELA's weights, unlike EXP and ATT. It is striking that TREE outperforms ATT on the $STT$ and $HB$ despite being significantly weaker, indicating that it must compensate with larger adaptation to and synergy with its lower-skilled partner. The second hypothesis we rule out is whether LEELA is a particularly bad senior/brain due to its strength, and our focals are better at adapting to their hands/brains simply because they are weaker and more similar to them. Replacing our focals with MAIA as a senior/hand (the most similar, weakest possible senior/brain) refutes this idea (see last column). We note that, while ATT and EXP are weaker than LEELA, they still achieve non-trivial scores ($>10\%$) in regular chess versus LEELA, indicating they are still strong chess agents.

We have established our central result: Our focal agents are objectively weaker than standard state-of-the-art AI, but their compatibility with MAIA is more than sufficient to defeat LEELA in both collaborative frameworks. We now investigate the mechanisms of skill-compatibility.

### 4.2 MECHANISMS OF ACHIEVING SKILL-COMPATIBILITY IN $STT$

How do our agents achieve skill-compatibility? In this section, we answer this question by analyzing agent behavior at the individual move level. We define the *win probability loss* of a move, which measures the degree to which any given move is sub-optimal. Notice that any chess move is either optimal, meaning it preserves the win probability of the previous position (as evaluated by a strong engine such as LEELA), or is sub-optimal, meaning it degrades the agent's win probability by a certain amount. We will define the win probability loss of a move, or simply

Table 2: Average losses for agents in $STT$.

| Agent | $G_t$ | $G_e$ | $G_a$ | Agent | $G_t$ | $G_e$ | $G_a$ |
|---|---|---|---|---|---|---|---|
| LEELA | $1.15^{**}$ | $1.16^{**}$ | $1.17^{**}$ | MAIA$_L$ | $4.46^{*}$ | $4.21^{*}$ | $4.19^{**}$ |
| focal | $1.91^{**}$ | $1.37^{**}$ | $1.43^{**}$ | MAIA$_F$ | $3.57^{*}$ | $3.37^{**}$ | $3.77^{**}$ |
| $\Delta_{G_f}$(LEELA, focal,∗) | $-0.76^{**}$ | $-0.21^{**}$ | $-0.26^{**}$ | $\Delta_{G_f}$(MAIA$_L$, MAIA$_F$,∗) | $0.89^{*}$ | $0.84^{*}$ | $0.42^{*}$ |
| $\Delta_{G_f}$(team$_L$, team$_F$,∗) | $0.13^{*}$ | $0.63^{*}$ | $0.16^{**}$ | | | | |

∗∗ $\leq \pm 0.02$, ∗ $\leq \pm 0.04$, see appendix for full error ranges

Table 3: Different effects of seniors on juniors
in $STT$. Effects that are stronger (p<0.05) than that of the opposing senior are in bold.

| | $G_t$ | | $G_e$ | | $G_a$ | |
|---|---|---|---|---|---|---|
| Agent | LEELA | TREE | LEELA | EXP | LEELA | ATT |
| Tricking : | $-0.03^{**}$ | $\mathbf{0.54^{**}}$ | $-0.27^{**}$ | $\mathbf{1.38^{**}}$ | $-0.01^{**}$ | $\mathbf{0.25^{**}}$ |
| Helping (Interceding Junior): | $0.26^{*}$ | $0.36^{*}$ | $0.30^{**}$ | $\mathbf{0.61^{**}}$ | $0.34^{**}$ | $0.21^{**}$ |
| Helping (Interceding Senior): | $0.15^{*}$ | $\mathbf{0.46^{*}}$ | $0.12^{*}$ | $\mathbf{1.88^{**}}$ | $0.20^{**}$ | $0.25^{**}$ |
| Indirect: | | $\mathbf{0.56^{*}}$ | | $-0.18^{**}$ | | $\mathbf{0.37^{**}}$ |

∗∗ $\leq \pm 0.07$, ∗ $\leq \pm 0.11$, see appendix for full error ranges

the *loss*, as the difference in the win probability of the board following the move to the win probability of the board preceding it. It ranges from 0 to 100. Given an agent $A$ and a condition $C$ on moves played by $A$ from a set of games $G_f$, we define a mean value $L_{G_f}(A,C)$ as follows: $L_{G_f}(A,C) = \frac{1}{|S|}\sum_{m \in S}\text{Loss}(m)$ where $S = \{m \in G_f | m$ satisfies $C$ and $m$ is played by $A\}$. To compare losses between agents $A_1$ and $A_2$, we define $\Delta_{G_f}(A_1,A_2,C) = L_{G_f}(A_1,C) - L_{G_f}(A_2,C)$, where setting $C = *$ means taking all moves.

For this section, we will be referring to agents as they appear in the tuple $\left\langle \begin{smallmatrix} \text{focal} \\ \text{MAIA}_F \end{smallmatrix} \middle| \begin{smallmatrix} \text{LEELA} \\ \text{MAIA}_L \end{smallmatrix} \right\rangle$, where focal refers to one of TREE, EXP$_T$, or ATT$_T$; LEELA denotes LEELA playing as the alter; MAIA$_F$ refers to the Focal team's junior MAIA agent, and MAIA$_L$ refers to the Alter team's junior MAIA agent. Note that MAIA$_F$ and MAIA$_L$ are both MAIA1100 agents. We will refer to this tuple as $G_f$ for simplicity, with $f$ being the starting letter of the corresponding focal. All agents and games in this section are in $STT$, so we omit the specifying subscript.

### 4.2.1 GENERAL EFFECTS ON JUNIOR PERFORMANCE

Table 2 shows the average loss by the agents involved at every move from the games played in STT. For all focal agents, we have $L_{G_f}(\text{focal},*) > L_{G_f}(\text{LEELA},*)$, yet $L_{G_f}(\text{MAIA}_F,*) < L_{G_f}(\text{MAIA}_L,*)$. This is a more granular, move-level statement of our central result: our focal engines sacrifice some optimality for the ability to influence and be skill-compatible with the sub-optimal agents they are interacting with.

Note also that $L_{G_t}(\text{TREE},*) > L_{G_a}(\text{ATT},*)$, yet $\Delta_{G_t}(\text{MAIA}_L,\text{MAIA}_F,*) > \Delta_{G_a}(\text{MAIA}_L,\text{MAIA}_F,*)$, implying TREE's results are more dependent on influencing the juniors present in the game. EXP combines low absolute $\Delta_{G_e}(\text{focal},\text{LEELA},*)$ with higher $\Delta_{G_e}(\text{MAIA}_L,\text{MAIA}_F,*)$ to get the best overall team loss difference among the three agents, which explains its higher score in the main evaluations.

To compare how these findings vary with position strength, we plot $\Delta_{G_f}(\text{MAIA}_L,\text{MAIA}_F,i)$, where $i$ stipulates the moves originate from boards where the probability of winning is equal to $i$ (Figure 4). For all focal engines, the gap is increasing in $i$: the closer the board is to winning, the more MAIA$_F$ outperforms MAIA$_L$. Winning boards are thus more critical, where the junior's moves have a chance to throw the game, compared to losing situations where the junior cannot bring about large positive changes to the evaluation.

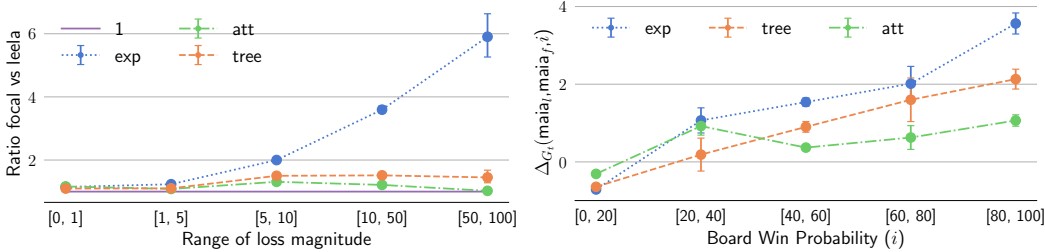

Figure 3: Ratio of probability of excess loss induction over different loss magnitudes

Figure 4: $\Delta_{G_f}(\text{MAIA}_L, \text{MAIA}_F)$ as a function of different board win probabilities

#### 4.2.2 Tricking, Helping, and Indirect Effects in $STT$

Having established that our focal agents induce a gap between $\text{MAIA}_F$ and $\text{MAIA}_L$ performance, we investigate three possible mechanisms by which they can achieve this: an immediate "tricking" effect, which we define as an induction of loss in $\text{MAIA}_L$ over a 1-ply horizon (the alter junior's next move), an immediate "helping" effect, which we define as reduction of loss in $\text{MAIA}_F$ over a 2-ply horizon (the focal junior's next move), and finally an indirect effect that cannot be measured over these short horizons (for example, early choices that impact how the game unfolds many moves into the future).

The chief difficulty of comparing the focals' effect on the juniors to that of leela on the juniors in the games played is that direct comparison would be potentially confounded by the different board distributions the opposing teams faced. We deal with the different distributions using two distinct methods detailed in the appendix. We present in table 3 the main results derived from the first method. Bold values indicates that the senior exhibits the effect in question. All agents display some tricking effect, with the magnitude being largest for EXP (1.38) and smallest for ATT (0.25). We do not observe a helping effect for ATT, but do so for the other focals. In particular, the helping effect for EXP is dramatically higher the when opponent leela intercedes (1.88 vs 0.61), and we believe this is due to the agent's preference for tricking the opponent junior when it intercedes instead of helping its own junior down the horizon. Finally, EXP has no beneficial indirect effect on the juniors (meaning longer than its optimization horizon of 2 plies), whereas the TREE and ATT have a measurable indirect effect even when they do not precede the juniors. Additionally, extra analysis in the supplement suggests that this indirect effect is more important than the tricking effect for these engines. These results demonstrate that there are multiple ways to influence the juniors, and the strongest agent, EXP, distinguishes itself by a complete lack of a beneficial indirect effect in favour of strong immediate effects, both helping and tricking.

### 4.3 Mechanisms of achieving skill-compatibility in $HB$

We now turn to the $HB$ framework. Note that $G_f$ now refers to the tuple $\left\{ \begin{smallmatrix} \text{focal} \\ \text{MAIA}_F \end{smallmatrix} \middle| \begin{smallmatrix} \text{LEELA} \\ \text{MAIA}_L \end{smallmatrix} \right\}$, and agents without subscript will be referring to those created for the $HB$ framework. We examine the effect of our focal agents (playing as brain) on the MAIA agents (playing as hand), and focus on two mechanisms: *intra-team* effects, where the brain picks a piece that causes MAIA to pick a better/worse move than it would have without interference, and *inter-team* effects, where the teams affect each other.

We first examine intra-team effects by computing the *savings*, the difference between the loss of the team's actual move played to the loss of the move MAIA would have selected without interference from the brain agent. Interestingly, Table 4 shows that EXP is the only brain exhibiting such an effect (0.3%), which is natural as it has been explicitly instructed to minimize the expected loss of its hand. TREE, which is the best performing agent in the framework, actually has negative savings (-0.2%), meaning its influence is causing its own hand to play worse moves. To analyze results more closely, we inspect the interaction between hands and brains.

There are four key hand-brain interactions: *agreement* (same move chosen), *blindsiding* (different moves but same piece-type, allowing the hand's move), *correction* (hand resamples to

Table 4: Comparison of team loss to hypothetical MAIA loss without brain in $HB$. Better in bold.

| | $G_t$ | | $G_e$ | | $G_a$ | | MAIA as focal | |
|---|---|---|---|---|---|---|---|---|
| | $\binom{\text{LEELA}}{\text{MAIA}_L}$ | $\binom{\text{TREE}}{\text{MAIA}_F}$ | $\binom{\text{LEELA}}{\text{MAIA}_L}$ | $\binom{\text{EXP}}{\text{MAIA}_F}$ | $\binom{\text{LEELA}}{\text{MAIA}_L}$ | $\binom{\text{ATT}}{\text{MAIA}_F}$ | $\binom{\text{LEELA}}{\text{MAIA}_L}$ | $\binom{\text{MAIA}}{\text{MAIA}_F}$ |
| True loss | 3.76** | **3.55**\** | 3.44** | **3.33**\** | 3.61** | **3.50**\** | **3.73**\* | 4.25* |
| MAIA loss | 3.75** | **3.29**\** | **3.40**\** | 3.62** | **3.59**\** | **3.43**\** | **3.39**\* | 3.60* |
| Savings | **-0.01**\** | -0.22** | -0.04** | **0.3**\** | -0.02** | -0.07** | **-0.32**\* | -0.65* |

** $\leq \pm 0.03$, * $\leq \pm 0.07$, see appendix for full error ranges

match the brain's different piece-type move), and *disagreement* (hand selects a different move after forced resampling). Detailed results on the proportions of each are in the supplement.

For all brains, the correction case yields some savings (4%–6%), and the disagreement yields a drop in the performance of the hands (1%–2%). Furthermore, the savings in the correction case are lower for TREE and ATT than they are for LEELA. TREE's effect does not appear to come from savings, but rather, we see that when TREE corrects its hand, it induces a high loss in the opponents' next move compared to LEELA's correction (4.8% vs 3.8%), showing a tricking action exerted by the TREE on the opponent team.

As TREE and ATT both increase their agreement with MAIA, we test the strategy of maximizing agreement and eliminating disagreement (as well as correction and blindsiding) entirely by letting MAIA play on its own against LEELA as the alter senior and MAIA as the alter junior, and it loses to LEELA with a 40% ±1.5 score. This confirms that disagreement-induced loss is more than compensated for by the benefit of correction of a strong brain.

### 4.4 EXPLORING IMPERFECT PARTNER MODELLING

All our experiments thus far have explored the results of senior agents designed to play with MAIA1100 (meaning a generic maia trained on many 1100 rated player games), tested on MAIA1100. Note that the term "designed for junior X" means, for ATT: that it trains with junior X; for TREE: that it runs MCTS using X to guide the search tree; and for EXP: that it uses junior X to simulate the next couple of moves. Now we investigate *cross-compatibility*, meaning whether seniors designed for these generic MAIA juniors are compatible with juniors that they are not explicitly designed for. We investigate three possible instantiations of cross-compatibility: cross-skill compatibility, specific player compatibility, and cross-style compatibility, and observe some success with the first two and an inability to generalize to radically different juniors.

#### 4.4.1 CROSS-SKILL COMPATIBILITY

Here, we use MAIA1900, the strongest available version of MAIA trained on data of 1900 rated players, as an alternative junior/hand. We analogously create TREE, EXP, and ATT agents designed to be compatible with MAIA1900. In this section, focalx denotes a focal designed for compatibility with MAIA1x00. Testing focal9 agents with MAIA1900 as the junior in 8 shows that they are able to beat LEELA in the frameworks, with the exception of TREE9 producing no gains in $STT$. We test cross-compatibility of the focal1 and focal9 agents by partnering them with each other's juniors. We emphasize that focal9 agents are not exposed to MAIA1100 prior to testing, and vice versa. As shown in Table 8 (appendix), focal9 agents are always compatible with MAIA1100 as a junior, irrespective of focal and framework, while the same is not always true of focal1 agents with MAIA1900. We hypothesize the assymetry is due to focal1 agents being more aggressive in playing suboptimally in order to exploit the junior, which backfires when MAIA1900 does not fall for exploits, whereas focal9 agents do not get explicitly punished if they over-conservatively fail to setup a trap for MAIA1100.

#### 4.4.2 SPECIFIC PLAYER COMPATIBILITY IN $STT$

We turn to validating whether the focal agents we created here using the generic MAIA models are compatible with individualized engines that are fine-tuned to mimic particular human players McIlroy-Young *et al.* (2020). We do so with the objective of exploring the applicability of

our method in the case where a complete model of the opponent/partner junior is not available (as is the case in most situations). Our initial experiment consists of individualized models of 2 players (rated 1650 and 1950), for which we designed specialized seniors, and compared the performance of these specialized seniors to that of seniors designed for generic MAIA1900. In these games, generally, generic focal agents win against leela, with a proportionally smaller margin than when seniors designed for generic/specific juniors are matched with the junior it was designed for. We now turn to more extensive player generalization testing.

Accordingly, we selected a random subset of 23 models, each trained on a particular human Lichess players rated between 1400 (43rd percentile skill) and 1900 (83rd percentile skill) as the junior for this experiment. We then use an EXP that internally uses the generic Maia agent with the nearest rating to the player in question, and have it play in $STT$ with that player's model. The median score of the different EXP agents over in this scenario with the 23 different juniors is 53.3% ($\pm 1\%$). While this is below EXP's performance of 66.5% in table 1, it nonetheless demonstrates that a generic approximation is sufficient to encapsulate skill-compatibility with individuals. In fact, out of 23 different players tested, EXP was shown to be winning ($p<0.05$) in 18 of the cases after 3000 games played, with the experiments on the remaining 5 players not showing statistical significance to that level.

### 4.4.3 CROSS-STYLE COMPATIBILITY IN $STT$

We now investigate using a completely different junior based on a non-neural architecture. To do that, we calibrate a low-depth version of stockfish (we call it SFW), to play at the skill-level of MAIA1100, and comparison in the appendix demonstrates that it is very different from any agent used so far. Note that there is no TREE agent. It is seen that EXP is able to obtain results against SFW, but ATT is unable to. Generalization is nonexistent, with engines trained with MAIA1100 losing when testing with a SFW junior and vice versa. We attribute this to the lack of similarity between SFW and MAIA1100. This does indicate, that our agents' compatibility is not a function of merely skill, but also style alignment.

## 5 LIMITATIONS AND DISCUSSION

Our work proposes a methodology for creating powerful agents that are *skill-compatible* with weaker partners. Our key finding is that, in a complex decision making setup as chess, skill-compatibility is a qualitatively and quantitatively measurable attribute of agents distinct from raw ability on the underlying task. Our designed frameworks show that in situations where strong engines are required to collaborate with weak engines, playing strength alone is insufficient to achieve the best results; it is necessary to achieve compatibility, even at the cost of pure strength. Finally, our three methodologies, each distinct in design and method of operation, demonstrate that there are multiple viable techniques to create agents that achieve this form of compatibility, with different agents using different strategies in-game. Indeed, some strategies center on helping the weak engine make better moves should it assume control, while others explicitly disrupt the compatibility of the adversary forcing weak opponent agents into errors, which even a strong partner like LEELA is unable to mitigate.

Our work therefore is an empirical proof-of-concept for skill-compatibility in chess, and provides a roadmap for the creation of human-compatible agents in this domain and beyond. While our paper does not speak directly to the prospect of skill-compatibility in other domains, we believe that a number of the techniques here are relatively general in nature, with clear analogues to other settings. For example, while the tree agent is very chess specific, and EXP is difficult to run in continuous environments, a number of ideas underlying these methodologies — EXP's short-term prediction, ATT's tandem training with the targeted skill — can be easily modified to fit different tasks, and offer potential for skill-compatibility in these tasks.

The training frameworks we propose use human-like MAIA agents as weak partners, and a natural next direction for future work is the design of experiments to test these methods with human chess players. Our scope likewise did not include instantiating the stronger partner beyond using LEELA, which offers opportunity to test robustness to modifications of the environment.

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

# 6 SUPPLEMENT

## 6.1 CODE RELEASE

Our code is released at `github.com/CSSLab/skill-compatibility-chess`. We also include several of our trained models.

## 6.2 METHODOLOGY DETAILS

### 6.2.1 TESTING AND ERROR RANGE COMPUTATION

To test a particular focal in $STT$, we run games of the form $\left\langle \begin{smallmatrix} focal \\ \text{MAIA} \end{smallmatrix} \middle| \begin{smallmatrix} \text{LEELA} \\ \text{MAIA} \end{smallmatrix} \right\rangle$. For any bitstring $s$, we play 2 games, with the focal and LEELA teams switching between black and white. This is done because some bitstrings are biased in favour of a particular color, and we therefore eliminate this bias by having each team playing both sides of the bitstring. Additionally, unfair bitstrings are undesirable, as it is likely that the team is less relevant than the color to achieve victory. Hence, they represent a source of unbiased noise to the result, by adding 1 win for each team. It is difficult to quantify fairness of a bitstring, (it is not sufficient to ensure equal number of senior and junior moves-the order matters). Therefore, for consistency, all experiments to test individual focal agents are sampling from the same set of bitstrings during testing, eliminating the possibility that some agents sample more unfair (and hence, noise-contributing) bitstrings during their testing.

No analogous measures are applicable for $HB$, as stochasticity is internal to the teams rather than being a characteristic of a game.

For both frameworks, testing consists of between 1000 and 10000 games, depending on our targeted significance. Here we detail computation of the standard error $se$ for the win-share displayed in the main section of the paper.

If we play $n$ games, with $W$ wins and $L$ losses, we can write the empirical win-share as

$$0.5 + \frac{\hat{w} - \hat{l}}{2}$$

where $\hat{w}$ and $\hat{l}$ are the empirical unbiased estimators of the true probabilities $w, l$, computed as $W/n$ and $L/n$ respectively. Since $w$, $l$, and $d$ (probability of a draw) form a trinomial distribution, we have

$$V(\hat{w} - \hat{l}) = \frac{w(1-w)}{n} + \frac{l(1-l)}{n} + \frac{2wl}{n}$$

which simplifies to

$$\frac{w + l - (w - l)^2}{n}$$

Plugging in the variance to get the $se$ of the win-share, we have

$$se = 0.5 \sqrt{\frac{w + l - (w - l)^2}{n}}$$

This is maximized with $w = l = 0.5$, and we have

$$se \leq \frac{0.5}{\sqrt{n}}$$

Our data is presented in percentages, so, this implies setting $n = 10000$ we guarantee $se <= 0.5\%$, and $n = 1000$ guarantees $se <= 1.5\%$. In practice, we are often able to get lower errors with lower $n$ because $w \neq l$.

All other quantities in our paper are sample means, with large sample sizes allowing the central limit theorem to be used to obtain their standard errors.

### 6.2.2 LEELA

We use the a 128x10-t60-2-5300 LEELA network, obtained from Vieri[1], with a 1500 node search. Against stockfish 13 (60k nodes), a strong classical engine that uses alpha-beta search, this version of LEELA obtains a score of $59 \pm 3$. Meloni[2] benchmarks stockfish 13 to human elo, so we deduce that our version of LEELA plays at around 3050 elo. While LEELA can be made significantly stronger if more nodes are used in search, limiting it to 1500 allows us to generate more games and run more tests, while still retaining superhuman capability. We use 1500 nodes for all seniors, for consistency. To compute the win probabilities of boards, needed to conduct most of our loss analysis, we use a separate instantiation of LEELA with the same parameters. In practice though, the LEELA used for evaluation is stronger than the LEELA playing as senior, because evaluations occur multiple times per move for different statistics, which, due to caching, is equivalent to working with more nodes.

### 6.2.3 ATT

To create ATT, a dataset of 10000 games (80% train, 10% validate, and 10% test) is generated of the following game $\left\langle \begin{smallmatrix} \text{LEELA} \\ \text{MAIA} \end{smallmatrix} \middle| \begin{smallmatrix} \text{LEELA} \\ \text{MAIA} \end{smallmatrix} \right\rangle$ for $STT$ or $\left\{ \begin{smallmatrix} \text{LEELA} \\ \text{MAIA} \end{smallmatrix} \middle| \begin{smallmatrix} \text{LEELA} \\ \text{MAIA} \end{smallmatrix} \right\}$ for $HB$. Then, starting with LEELA's weights, and using a learning rate of $10^{-5}$, and 10000 iterations, we run back-propagation to update LEELA's policy and value neural network. We use the version of MAIA for which we are attempting to achieve compatibility in this training scheme.

As this is a tuning task, and we are starting with LEELA weights, we perform some parameter tuning, with the objective not necessarily to find optimal parameters, but rather to find a set thereof that compromises between cost and robustness. The main time cost comes from generating the datasets of games, and training the agents.

To do so, we train models with learning rates from $10^{-1}$ to $10^{-6}$ on dataset sizes of 1000, 10000, and 100000 games, and a short training time of 1000 iterations. We test these models with a small number of matches that is sufficient to determine whether the training procedure produced viable engines (viable simply means plausible, not necessarily successful). See table 5. Some learning rates catastrophically fail and exhibit nonsensical learning curves, or produce agents that lose a large majority of their games.

From the above, we decide to use 10000 games as our dataset size, as it appears that a range of learning rates are viable on it, and it is not as costly to generate as 100000. We also settle on learning rate of $10^{-5}$, as this rate is more robust in small datasets than $10^{-4}$ and requires less time to convergence than $10^{-6}$.

We note that there appears to be a connection between the quality of the policy and value accuracy curves and actual performance, meaning, models that overfit or fail to converge also perform poorly when testing on the frameworks. Accordingly, we observe that 100000 iterations produces more complete convergence curves for this particular learning rate and dataset size choice.

After having selected hyper-parameters which we deem plausible, we run our first full training run 8 times to ensure that performance on the framework following training is repeatable, rather than a product of chance. The worst model of the 8 has a win-rate of 53.5 $\pm$1, and the best model a win-rate of 56.0 $\pm$1.

Note that our goal is to find valid, stable hyper-parameters, which does not preclude the existence of better sets of hyper-parameters. For all other models, (meaning different frameworks, or different MAIA juniors), we use these parameters, and if convergence issues arrive, we modify parameters heuristically. This type of modification was not actually needed for any models in the main paper, but was required for a few additional models which we detail here.

The rest of the hyper-parameters can be found in the configuration files in the linked code.

---

[1]https://lczero.org/dev/wiki/best-nets-for-lc0

[2]https://www.melonimarco.it/en/2021/03/08/stockfish-and-lc0-test-at-different-number-of-nodes/

### 6.2.4 EXP

EXP is more expensive to run than the standard neural netework engines as it makes multiple calls to engines as subroutines to compute the move that maximizes expectation. Accordingly, it is unfeasible to do a full search over all legal moves, as that may consume up to hundreds of times as much compute. Likewise, these engines use shallow LEELA engines for board evaluation to determine their move selection. For $STT$, we compute the expectation over the top 5 moves, with evaluation conducted at 300 nodes, whereas in $HB$, for each piece, we compute the expectation over the top 3 MAIA moves (meaning 18 moves checked total), with evaluation conducted at 50 nodes.

### 6.2.5 DETAILING METHOD 1 USED IN 4.2.3

While involved, the technique used here comes with the added benefit that no engines play additional moves outside what has already played within evaluation games, and therefore the comparison is more pertinent to the games themselves. To study these three mechanisms, we analyze special sequences on the board. Since in $STT$, agent selection is done via coin flipping, games can be represented as a bitstring, $s$ where 1s indicate senior moves and 0s indicate junior moves. We now detail how we compute the values present in table 3. By computing the loss on moves that are preceded by specific substrings, we can isolate different effects. Accordingly, we define $L_{G_f}(A,s)$ where $s$ is a condition that stipulates that the moves must be preceded by a (partial) bitstring $s$ in $G_f$. For example, $L_{G_f}(\text{MAIA}_F,1)$ means that we are computing the loss of MAIA$_F$ only when its move comes immediately after LEELA's, whereas $L_{G_f}(\text{MAIA},0)$ means the moves come after MAIA$_L$'s.

As mentioned earlier, in order to compare the tricking effect of a senior $S_1$ to a senior $S_2$, we do not directly compare $L_{G_f}(J_2,1)$ (measuring how $S_1$ tricks $J_2$) to $L_{G_f}(J_1,1)$ (measuring how $S_2$ tricks $J_1$), as $J_1$ and $J_2$ are playing on different board distributions (the distributions of the board of the two teams are not the same, the simplest example being that the focal team will have more winning boards). We perform an indirect comparison of each senior's effect to its own junior's effect, as they share a board distribution, and all necessary moves are already present. We do the same for both seniors and then compare these quantities. Formally, we define $I_{G_f}(A,s) = L_{G_f}(A,1 \oplus s) - L_{G_f}(A,0 \oplus s)$ where $\oplus$ is concatenation, the 1 and 0 denote the comparison of the loss induced by the senior being present in that position to that by the junior, and $s$ is a string to standardize the agents that play in between should we be measuring an effect across more than 1 ply.

The immediate tricking effect of a senior $S$ under this definition can thus be computed as $I_{G_f}(J_{opp},\phi)$ where $\phi$ is the empty string and $J_{opp}$ is the opposing junior.

To measure the immediate helping effect of a senior $S$ on its partner junior $J_{par}$, note that the opposing team must play a move prior to $J_{par}$, and there are two possible situations depending on which opponent plays. There are thus two helping effects to be measured, $I_{G_f}(J_{par},1)$ and $I_{G_f}(J_{par},0)$, denoting the opponent senior and junior being the interceding agents, respectively.

An in order to examine the indirect effect, to see whether the focal agents affect the performance of their juniors over longer time horizons. We compute this as $\Delta_{G_f}(\text{MAIA}_L,\text{MAIA}_F,00)$, where two zeros indicate the move must not be preceded by any senior for 2 plies.

### 6.2.6 DETAILING METHOD 2 USED IN 4.2.3

We perform this analysis with LEELA and each focal agent on two separate board distributions: the set of boards that LEELA's team and that the focal's team encountered in-game. Table 6 shows these results for different focal agents. EXP induces similar loss in MAIA regardless of the board distribution ($4.47 \approx 4.39$). This is in line with EXP's myopic optimization objective. The gulf between EXP and LEELA remains large regardless if we compare the distributions as seen in game ($4.39 > 3.25$) or if we standerde the board distribution ($4.39 > 3.65$ and $4.47 > 3.25$). In contrast, there is a notable degradation in the loss induction abilities of TREE and ATT when eliminating distributional effects ($3.89 < 4.85$ and $3.95 < 4.46$). Consequently, the standardized comparisons for these focals to LEELA shows a much closer result than the unstandardized in-game observations. Interestingly, comparing along the minor diagonal shows that LEELA induces *more* loss than these two agents ($4.50 > 3.89$ and $4.45 > 3.87$), suggesting that the distributional effects induced by TREE and ATT are actually more important than their direct effects.

### 6.3 HARDWARE

We made use of four Tesla K80 GPU's for the purpose of experimentation, each with a VRAM of 12 GB. We show in table 11 the times taken for the most important tasks of the paper. As an example, if we wish to generate the games for an ATT$_T$ agent, train it, test it to $se=0.5$, and obtain analysis metrics, it would take 10 hours for generation, 3 hours for training, 10 hours for testing, and 12 hours for analysis, a total of 35 hours. Alongside experiments that are not included in the paper, we approximate the total compute time used by the project to be approximately 1 year's worth of our GPUs.

### 6.4 EXTRA EXPERIMENTS

#### 6.4.1 AGREEMENT RATE OF VARIOUS ENGINES

We compute the agreement rate of various engines used in our experiments in figure 5. Notice how the ATT agents from both frameworks and calibrated to both juniors all play similar moves to each other and to LEELA, on which they are based. MAIA1100 and MAIA1900 also exhibit similarity to each other and dissimilarity to ATT agents. TREE exhibits moderate similarity to both the MAIA agents and the LEELA-derived agents. SFW, a weakened version (25 nodes) of stockfish 8 used in a later experiment, is entirely dissimilar to any other agent, likely due to its architectural uniqueness.

#### 6.4.2 MODIFICATION OF TRAINING TARGET FOR ATT IN $STT$

The training procedure for ATT includes back-propagating on moves that both LEELA and MAIA play in $STT$. Semantically, this updates the value-head to take into account MAIA's interventions, however, it also updates the policy-head to learn MAIA's moves, which we believed would weaken the engine. Therefore, we created a version of the engine that only learns LEELA's moves, as to not affect the policy with MAIA's moves, and only to have the value-head change to adapt to MAIA's interventions. We also use an increased training games of 40000 because convergence with smaller datasets was worse (exclusion of MAIA moves halves quantity of data, and reduces its diversity, making it prone to overfitting). Interestingly, this turns out to be less effective than the default version in $STT$. ($52\%\pm0.5 < 55.0\%\pm0.5$) than including the MAIA policy moves, although the engine turns out to be very strong in raw strength, achieving a $36.5\%\pm1.5 > 14.0\%\pm1.5$. It is expected that the engine is stronger, as it is not ingesting MAIA moves, and we suspect that in $STT$, training the policy on MAIA moves is actually beneficial, as it allows the agent to conduct search at least partially based on MAIA's moves, mimicking TREE to an extent.

#### 6.4.3 COMPARISON OF TRICKING VS HELPING STRATEGIES IN $STT$

In order to compare the effect of attempting purely to sabotage MAIA$_L$, to that of purely aiding the MAIA$_F$, we modify the algorithm of EXP to $m = \underset{m}{\operatorname{argmax}}\big(\mathbb{E}_{s\in\{0,1\}}[w|(m,s)]\big)$, which effectively optimizes only one ply in the future, eliminating any helping effect which requires at least 2 ply foresight and making this a pure tricking engine. In order to isolate the helping effect, we use $m = \underset{m}{\operatorname{argmax}}\big(\mathbb{E}_{s\in\{10,11\}}[w|(m,s)]\big)$, which (falsely) assumes an un-exploitable LEELA is always playing the opponent move, thereby forcing the optimization to be solely to help MAIA$_F$. Both versions are able to beat LEELA in the framework, however the tricking version achieves a higher score of $62.5\%\pm1$ as opposed to the helping version which achives a score of $57.0\%\pm1$, both lower than that achieved by the actual EXP. We suspect that pure tricking is easier to conduct than pure helping, as there is no interceding agent to account for, and a shorter time horizon, hence less branching.

## 6.5 Extra Figures and Tables

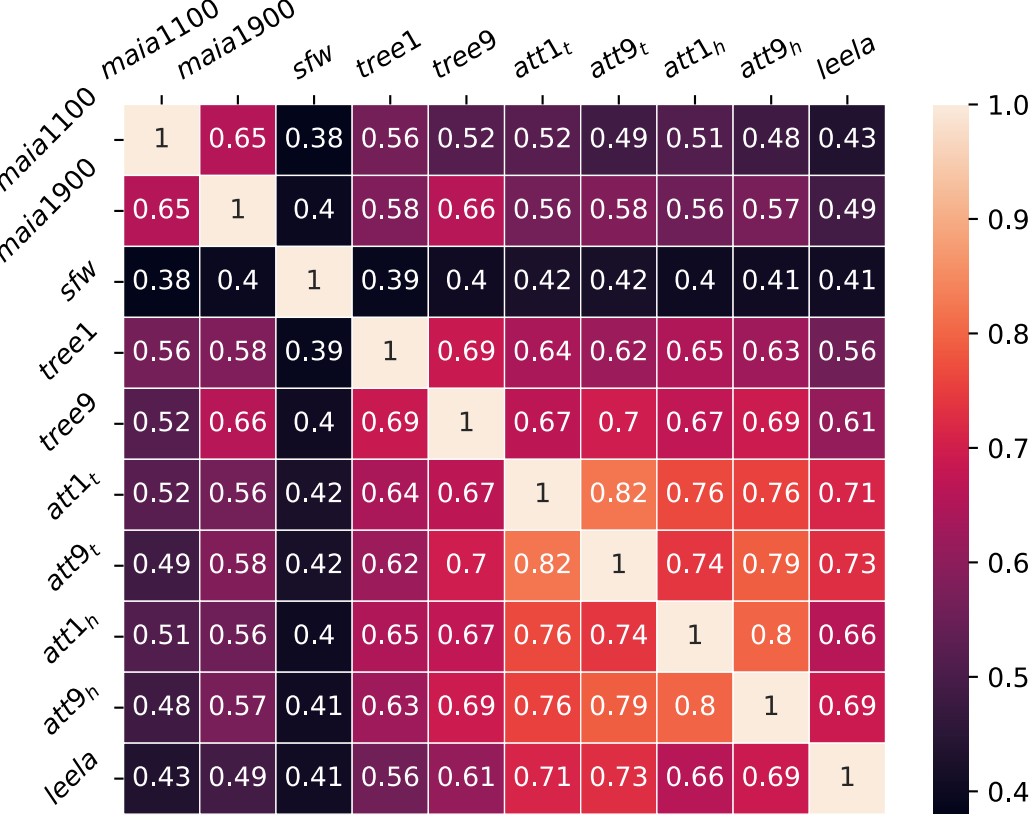

Figure 5: Agreement rate of different engines with each other

Table 5: Viability of $\textsc{att}_\textsc{t}$ engines created according to learning rate and dataset size

| Learning Rate | $10^{-1}$ | $10^{-2}$ | $10^{-3}$ | $10^{-4}$ | $10^{-5}$ | $10^{-6}$ |
|---|---|---|---|---|---|---|
| 1000 Games | X | X | X | X | ✓ | ✓ |
| 10000 Games | X | X | X | ✓ | ✓ | ✓ |
| 100000 Games | X | X | ✓ | ✓ | ✓ | ✓ |

## 6.6 Main Paper Tables with Standard Deviation

Tables 12-15, show the standard deviation of each value from the main text.

Table 6:
MAIA loss induced by different seniors in distributions occurring to different teams in $STT$

| MAIA loss induced by | $G_t$ | | $G_e$ | | $G_a$ | |
|---|---|---|---|---|---|---|
| | LEELA | TREE | LEELA | EXP | LEELA | ATT |
| Distribution of $\binom{\text{LEELA}}{\text{MAIA}_\text{L}}$ | 3.60±0.02 | 3.89±0.02 | 3.25±0.02 | 4.47 ±0.02 | 3.87±0.02 | 3.95±0.02 |
| Distribution of $\binom{\text{FOCAL}}{\text{MAIA}_\text{F}}$ | 4.50±0.03 | 4.85±0.03 | 3.65±0.02 | 4.39±0.02 | 4.33±0.01 | 4.46±0.01 |

Table 7: Metrics by interaction type for TREE and ATT in $HB$

| Interaction | Agreement | | Correction | | Disagreement | | Blindsiding | |
|---|---|---|---|---|---|---|---|---|
| Team | $\binom{\text{LEELA}}{\text{MAIA}_\text{L}}$ | $\binom{\text{FOCAL}}{\text{MAIA}_\text{F}}$ | $\binom{\text{LEELA}}{\text{MAIA}_\text{L}}$ | $\binom{\text{FOCAL}}{\text{MAIA}_\text{F}}$ | $\binom{\text{LEELA}}{\text{MAIA}_\text{L}}$ | $\binom{\text{FOCAL}}{\text{MAIA}_\text{F}}$ | $\binom{\text{LEELA}}{\text{MAIA}_\text{L}}$ | $\binom{\text{FOCAL}}{\text{MAIA}_\text{F}}$ |
| $G_t$ Distribution | 44±1 | 60±1 | **20**±1 | 16±1 | 21±1 | **14**±1 | 15±1 | 10±1 |
| $G_t$ Savings | 0 | 0 | **5.4**±0.1 | 4.3±0.1 | -2.0±0.1 | -2.1±0.1 | 0 | 0 |
| $G_t$ Opponent loss | 3.2±0.1 | 3.4±0.1 | 3.8±0.1 | **4.8**±0.1 | 4.0±0.1 | **4.4**±0.1 | 3.3 ±0.1 | **3.8**±0.1 |
| $G_a$ Distribution | 44±1 | 52±1 | 20±1 | 20±1 | 20±1 | **16**±1 | 16±1 | 11±1 |
| $G_a$ Savings | 0 | 0 | **5.2**±0.1 | 4.6±0.1 | -1.8±0.1 | -1.9±0.1 | 0 | 0 |
| $G_a$ Opponent loss | 3.2±0.1 | 3.2±0.1 | 3.8±0.1 | 4.0±0.1 | 3.9±0.1 | 3.9±0.1 | 3.3 ±0.1 | **3.7**±0.1 |

Table 8: Generalization results with MAIA1100 and MAIA1900 as junior partners.

| | $STT$ framework | | | | | |
|---|---|---|---|---|---|---|
| | TREE | | EXP$_\text{T}$ | | ATT$_\text{T}$ | |
| Tested on junior | MAIA1100 | MAIA1900 | MAIA1100 | MAIA1900 | MAIA1100 | MAIA1900 |
| Designed for MAIA1100 | 56.5±0.5 | 41.5±1.5 | 66.5±0.5 | 53.0 ±0.5 | 55.0±0.5 | 51.5±0.5 |
| Designed for MAIA1900 | 51.0±0.5 | 50.0±0.5 | 55.0±0.5 | 65.5±1.0 | 52.0±0.5 | 53.0±0.5 |
| | $HB$ framework | | | | | |
| | TREE | | EXP$_\text{H}$ | | ATT$_\text{H}$ | |
| Tested on junior | MAIA1100 | MAIA1900 | MAIA1100 | MAIA1900 | MAIA1100 | MAIA1900 |
| Designed for MAIA1100 | 60.0±0.5 | 52.5±0.5 | 55.0±0.5 | 45.5 ±1.5 | 56.0±0.5 | 37.5±1.5 |
| Designed for MAIA1900 | 57.0±0.5 | 58.0±0.5 | 51.5±0.5 | 55.0±0.5 | 54.0±0.5 | 54.0±0.5 |

Table 9: Generalization results with MAIA1100 and SFW on $STT$

| Tested on | EXP$_t$ | | ATT$_t$ | |
|---|---|---|---|---|
| | MAIA1100 | sfw | MAIA1100 | sfw |
| Designed for MAIA1100 | 66.5±0.5 | 44.0±1.0 | 55.0±0.5 | 44.0±1.0 |
| Designed for sfw | 43.5±1.0 | 55.0±0.5 | 48.5±0.5 | 51.0±0.5 |

Table 10: Generalization results with Specific Players in $STT$

| Player A (1650 rating) | | | |
|---|---|---|---|
| Senior trained for | Att | Tree | Exp |
| Maia1900 | 51 ±0.5 | 52.5 ±0.5 | 57 ±2 |
| Player A | Not trained | 51.5 ±0.5 | 68 ±2 |
| Player B (1950 rating) | | | |
| Type of Senior | Att | Tree | Exp |
| Maia1900 | 51 ±0.5 | 52 ±0.5 | 53 ±2 |
| Player B | Not trained | 46.5 ±0.5 | 66.5 ±2 |

Table 11: Approximate times of main tasks involved in experimentation

| Task | Approximate Time (h) |
|---|---|
| 1000 $STT$ games, no EXP | 1 |
| 1000 $STT$ games, with EXP | 2-3 |
| 1000 $HB$ games, no EXP | 2 |
| 1000 $HB$ games, with EXP | 2-3 |
| 1000 games with metric collection | 4 |
| 10000 training iterations | 3 |

Table 12: Table 1 with standard deviations

| | $STT$ | | | | $HB$ | | | |
|---|---|---|---|---|---|---|---|---|
| Game Setup | TREE | $EXP_T$ | $ATT_T$ | MAIA | TREE | $EXP_H$ | $ATT_H$ | MAIA |
| focal LEELA / MAIA MAIA | 56.5±0.5 | 66.5±0.5 | 55.0±0.5 | 7.5 ±1.5 | 60.0±0.5 | 56.5±0.5 | 55.0±0.5 | 27.5 ±1.5 |
| focal / LEELA | 0.5±0.5 | 14.0 ±1.5 | 12.5 ±1.5 | 0.0±0 | 0.5±0.5 | N/A | 4.5 ±1.5 | 0.0±0 |

Table 13: Table 2 with standard deviations

| Agent | $G_t$ | $G_e$ | $G_a$ | Agent | $G_t$ | $G_e$ | $G_a$ |
|---|---|---|---|---|---|---|---|
| LEELA | 1.15 ±0.01 | 1.16 ±0.01 | 1.17 ±0.01 | $MAIA_L$ | 4.46 ±0.04 | 4.21 ±0.04 | 4.19 ±0.02 |
| focal | 1.91 ±0.01 | 1.37 ±0.01 | 1.43 ±0.01 | $MAIA_F$ | 3.57 ±0.03 | 3.37 ±0.02 | 3.77 ±0.02 |
| $\Delta_{G_f}$(LEELA, focal,*) | -0.76 ±0.02 | -0.21 ±0.02 | -0.26 ±0.02 | $\Delta_{G_f}$($MAIA_L$, $MAIA_F$,*) | 0.89 ±0.04 | 0.84 ±0.04 | 0.42 ±0.03 |
| $\Delta_{G_f}$($team_l$, $team_f$,*) | 0.13 ±0.03 | 0.63 ±0.04 | 0.16 ±0.02 | | | | |

Table 14: Table 3 with standard deviations

| | $G_t$ | | $G_e$ | | $G_a$ | |
|---|---|---|---|---|---|---|
| Agent | LEELA | TREE | LEELA | EXP | LEELA | ATT |
| Tricking: $I(J_{opp},\phi)$ | -0.03±0.06 | **0.54**±0.07 | -0.27±0.04 | **1.38**±0.05 | -0.01±0.06 | **0.25**±0.04 |
| Helping: $I(J_{par},0)$ | 0.26±0.11 | 0.36±0.10 | 0.30±0.07 | **0.61**±0.07 | 0.34±0.06 | 0.21±0.05 |
| Helping: $I(J_{par},1)$ | 0.15±0.09 | **0.46**±0.09 | 0.12±0.09 | **1.88**±0.07 | 0.20±0.06 | 0.25±0.05 |
| Indirect: $\Delta$($MAIA_L$, $MAIA_F$,00) | **0.56**±0.10 | | **-0.18**±0.07 | | **0.37**±0.06 | |

Table 15: Table 4 with standard deviations

| | $G_t$ | | $G_e$ | | $G_a$ | | MAIA as focal | |
|---|---|---|---|---|---|---|---|---|
| | $\binom{LEELA}{MAIA_L}$ | $\binom{TREE}{MAIA_F}$ | $\binom{LEELA}{MAIA_L}$ | $\binom{EXP}{MAIA_F}$ | $\binom{LEELA}{MAIA_L}$ | $\binom{ATT}{MAIA_F}$ | $\binom{LEELA}{MAIA_L}$ | $\binom{MAIA}{MAIA_F}$ |
| True loss | 3.76±0.02 | **3.55**±0.02 | 3.44±0.02 | **3.33**±0.02 | 3.61±0.02 | **3.50**±0.02 | **3.73**±0.05 | 4.25±0.05 |
| MAIA loss | 3.75±0.02 | **3.29**±0.02 | **3.40**±0.02 | 3.62±0.02 | 3.59±0.02 | **3.43**±0.02 | **3.39**±0.05 | 3.60±0.05 |
| Savings | **-0.01**±0.03 | -0.22±0.03 | -0.04±0.03 | **0.3**±0.03 | -0.02±0.03 | -0.07±0.03 | **-0.32**±0.07 | -0.65±0.07 |

