# OpenReview forum: "Designing Skill-Compatible AI: Methodologies and Frameworks in Chess"
_ICLR.cc/2024/Conference — ICLR 2024 poster_

### Official Review · Reviewer_XZq8 · 2023-10-23

**Soundness:** 3 good
**Presentation:** 3 good
**Contribution:** 3 good
**Rating:** 8
**Confidence:** 4

**Summary:**

The paper studies the problem of AI agents acting in team settings, particularly where the AI agent is more capable at the task than it's partner(s). Chess with both sides played by two players is used as a study domain, with two formulations of team play considered:
1) stochastically switching between which of a pair of agents on the team chooses for any given move
2) having one agent choose the piece type and the other choose the specific piece and move to make

These formulations are meant to target the capabilities of the stronger agent to work with the weaker agent either due to temporal uncertainty (1) or only indirect/partial control on action selection (2).

Three agents are evaluated to play the role of the strong agent:
1) Tree agent: Uses Monte Carlo Tree Search (MCTS) to augment a baseline policy (agent)
2) Expector agent: Maximize win probability over a short horizon given access to models of all agents
3) Attuned agent: Fine-tunes the agent from traces of agent pairs playing

In all cases the models have full access to models of the other agents in the game. Evaluations demonstrate that all three agents proposed improve over a naive baseline where a strong and weak agent are paired. Comparisons are made in terms of win rate and change in win probability for moves. Detailed studies are made to examine the mechanisms for the improvements, along with a small study of the ability of the expector agent to generalize to imperfect partner models.

**Strengths:**

# originality
Framing collaboration (compatibility) as the ability to cope with partners under the specific cases here is novel. While AI teaming in chess is well-established, using stochastic turn taking or hand-brain teams as a way to assess agent coordination is novel. The proposed agents are all reasonable extensions of methods for opponent or partner modeling to these scenarios.

# quality
The experiments thoroughly examine the performance improvements of the varying approaches under two different conditions. The tests quantify uncertainty and demonstrate a clear benefit both to long-term win rates and intermediary probabilities of selecting good moves (optimal with respect to a given agents' predictions). These demonstrate that the models for agent pairing are achieving the desired effect of producing teams stronger than their individual composite agents.

# clarity
The text is clear on the technical approach and provides good framing of the core frameworks.

# significance
Human-AI collaboration is of growing interest (and public concern) as AI systems are more widely deployed. Developing new methods to assess how teams work together and can be robust to differing agent capabilities is a timely and important topic. This will be widely of interest across subcommunities at ICLR (and beyond).

Working specifically in games has clear significance to research in game playing, reinforcement learning, and efforts at multi-agent coordination.

**Weaknesses:**

# originality
No major comments. If there is room, the related work could make further reference to the literature on zero-shot coordination, which shares the goal of enabling teams of agents to work well together, but subject to the constraint of lacking a model of the partner agents. Some examples:

- Hu, Hengyuan, Adam Lerer, Alex Peysakhovich, and Jakob Foerster. ""other-play" for zero-shot coordination." In International Conference on Machine Learning, pp. 4399-4410. PMLR, 2020.
- Lupu, Andrei, Brandon Cui, Hengyuan Hu, and Jakob Foerster. "Trajectory diversity for zero-shot coordination." In International conference on machine learning, pp. 7204-7213. PMLR, 2021.
- Treutlein, J., Dennis, M., Oesterheld, C. and Foerster, J., 2021, July. A new formalism, method and open issues for zero-shot coordination. In International Conference on Machine Learning (pp. 10413-10423). PMLR.

# quality
Results on the mechanisms behind the improvement gained by different focal agents are murky. For example, "tricking" may be implicated for EXP, but not TREE or ATT (though I may have misunderstood the text on this point). In general, it is hard to assess how strong some of these results are and what the implications may be. More specific questions are provided below.

Conversely, few results test the generalization capabilities of the agents, particularly when relaxing the assumption of access to a perfect model of partners and opponents.

# clarity
The results presentation is somewhat murky, particularly in the detailed analysis of mechanisms (see the questions for specific details). In general the results would benefit from a clear statement of the outcomes of the analyses at the start of the section, followed by the existing detailed description of results. This will help readers glean the overall conclusions that get lost in some of the details.

# significance
The primary limitation to the impact of the results is the requirement for agents to have models of all relevant agents in the scenario. One experiment slightly relaxes this constraint and the results become substantially weaker. It would be beneficial to add some remarks on this point and future directions mitigate these weaknesses with imperfect partners. As a first work on this method I do not view these as substantial limitations, but they deserve some additional comment.

**Questions:**

- Tables 1 & 2
	- Minor suggestion: Combine these two tables into one with a row defining the task (STT vs HB).
	- If possible it would help to include the error ranges in the main text.
	- Is there any intuition for why TREE does so much better than EXP in HB? The improvement is the opposite of what is observed in STT. Is the problem that in HB EXP has too limited of a search depth to handle the relevant decision horizon?
- Section 4.2.2
	- The final conclusion from this analysis was unclear.
	- What are the implications that the standardized and non-standardized board position distributions differ substantially for TREE and ATT, but not EXP? This seems to suggest tricking is the not the main effect (as opposed to some more general distributional difference), but that was not completely clear either.
	- It may be better to remove this analysis (to the appendix) and provide more detail in the main text on the normalization applied in 4.2.3.
	- Section 4.2 would benefit from stating the conclusions drawn on tricking, helping, and indirect effects at the start of the section. A similar comment applies to section 4.3.
	- I could not understand from the text: which agents demonstrate tricking (and under what analysis)?
		- For helping it is more clear (based on the more complex analysis) what is happening.
- Section 4.4
	- Are there results when the MAIA agent is selected without a rating prior? This would be interesting to see to understand how well a focal can compensate when the "amount of weakness" of the partner is highly uncertain.
	- One approach would be randomizing the skill of the partner agents over a reasonable range of ratings.
	- Are there comparable results for using TREE? I understand ATT is more computationally demanding to add, but those results would also be interesting.
	- In general I consider these results to be some of the most important for the work and would trade less detailed analysis from sections 4.1, 4.2, and 4.3 for further elucidation of how well the approach generalizes when given imperfect partner information.
- On the supplement
	- The results on using MAIA1900 and stockfish are both good information to add to the main body. These provide more color to the discussion about capabilities to partner with other agents and would be welcome in the main text.
	- I personally would trade these for the space used in sections 4.1, 4.2, and 4.3 as the final conclusions on the mechanisms at play remain unclear to me, while the conclusions about capacity to generalize (or not) seem more clear.
	- "This does indicate, that our agents' compatibility is not a function of merely skill, but also style." An important point to make in the main text! Also very relevant for future work that would investigate other ways to enhance agent compatibility.

Two broad conceptual questions came to mind that I would appreciate further comments on:
- Why is inter-temporal coordination the right framing for evaluating agent compatibility?
	- Perhaps sketch some specific applications where this is a natural interaction paradigm outside chess. The example that came to mind is autonomous vehicles (and robotics more generally), where humans may take over at arbitrary points from an AI driver.
- How correlated is this notion of interpretable with prior efforts?
	- The term interpretable is quite loaded in the literature, so I am trying to understand why it is a good description of the method here.
	- Prior efforts normally aim to enable inspection of the behavior of a model, but that is not obviously what is demonstrated here. Weak agents can perform better when in scenarios prepared by the strong agents. But it's not obvious that is due to any ability to "interpret" these situations, as opposed to being given better initial states to use.

---

**Author discussion amendment**

I have revised my score up to reflect the improved paper structure and clearer promising results on generalization with other types of agents. These reflect the methods generalize across some dimensions (human players and skills) but not others (whatever makes Stockfish different to humans). These point toward important future research efforts.

---

> ### Author Response · Authors · 2023-11-19
>
> Response Part 1/2:
>
> We thank you for taking the time to provide a detailed review. We will attempt to address your questions (in addition to modifying our manuscript by adding emphasis on section 4.4 as described in our main response).
>
> Your comments on Tables 1 & 2:
>
> We thank you for the suggestion. We merged Tables 1 and 2 and agree that this has improved readability. We have left the error ranges in the appendix as they tend to clutter the tables and detract from the actual values.
> As for our speculation of why tree outperforms exp in hand and brain, we suspect that could potentially be due to the adversarial effect that tree can effect in HB. The exp agents in HB, with their horizon not extending beyond their hand’s moves, are the only agents actually able to reduce the loss of their hands. However, we expect that they will not be able to have an adversarial effect on the opponent hand, which might be quite important. That said, it is difficult to measure HB’s adversarial effect analogously because its interactions with the hands cannot be broken down into corrections and agreements due to it not outputting a full move.
>
> Your comments on Section 4.2.2:
>
> We have modified Section 4.2 to state the main takeaways more explicitly, and abstract the analysis to the appendix. The main takeaways from 4.2.2 and 4.2.3 are that all agents display some tricking effects (the expectors to the greatest extent). Expectors and tree exhibit a measurable helping effect. Finally, tree and attuned agents demonstrate a distributional effect. The analysis performed in 4.2.2 is less comprehensive (it neglects helping), but is significantly simpler than that of 4.2.3. The results of 4.2.2 and 4.2.3 show the same thing, except that 4.2.2 allows us to look at the interplay between the tricking effect and the distributional effect, whereas 4.2.3 gives us a clearer look at everything in isolation.
>
> To clarify 4.2.2, the experiment demonstrates that exp is not dependent on a distributional effect as far as observed by the fact that its trickiness (which is high) is almost unchanged when measured on Leela’s distribution and its own distribution. For the other agents, we observe that trickiness is modulated by the distribution (the ability of focals to trick is amplified on the distribution of the focals compared to on leela's distribution), as well as a greater inherent tricking effect than leela even when standardized for distribution. The final observation was that if we had att and tree play on leela's distribution, they were less tricky than leela playing on their distributions, suggesting that their distributional effect is, in a sense, responsible for a large part (but not all) of their trickiness.
>
> The "distributional effect" measured in 4.2.3 is slightly different than that in 4.2.2, it instead measures maia's performance when not preceded by any seniors (as opposed to how distributions modulate trickiness), and we see similarly that tree and att have a distributional effect there and exp does not.
>
> Your comments on Section 4.4:
>
> While we did not incorporate skill-uncertainty into our specific player compatibility experiments, we address the issue of skill-estimation in the supplement (1100 vs 1900). This does allow us to answer the question of how the results look when the amount of weakness is incorrectly estimated. Indeed, we find that seniors designed to play with stronger juniors fare better against weaker juniors than vice versa, and we suspect that overexploitation is punished whereas under-exploitation is not. This part was formerly in the appendix, and is now in the main body. We agree with you that testing att and tree is valuable on these agents, and we also understand the computational cost of the task. Accordingly, we add a previously omitted experiment where we run the other agents  on these two players. We also implement your suggestion of reducing the 4.2.3 and 4.2.2 in favor of expanding this section notably, and having it play a more important role in the paper, which is our most major implemented revision after reading all reviews.
>
> Your comments on the supplement:
>
> Tying to our expansion of cross-style compatibility, and as mentioned in the main response, we also promote the cross-skill compatibility and Stockfish tests to the main body of the paper. We agree that they represent an important avenue for improvement and further work.

---

> > ### Author Response · Authors · 2023-11-20
> >
> > Response Part 2/2:
> >
> > In response to the question concerning the inter-temporal nature of the definition of compatibility:
> >
> > While some prior works mentioned in our paper explore simultaneous settings, there are real life applications where the roles of humans and AI is temporally interleaved. STT can be likened to real life situations, such as autonomous vehicles with humans, where we have 2 agents of different abilities essentially playing the same role, with a lack of determinism concerning who is in the literal driver’s seat. In these situations, the “autopilot” must not put the car in a state where a human driver could not navigate safely, should they choose to disengage the “autopilot”. HB lends itself to asymmetrical roles contributing to a decision, with predictable roles for AI and humans. For example, automated stock screening tools are used to narrow down options prior to selection by humans. Here, we could formulate the human as an agent with a particular probability of selecting a particular stock if it is included in the suggestion. Decision-making in intertemporal symmetric and asymmetric systems is thus a mode of interaction between humans and AI agents worth investigating.
> >
> > In response to the question concerning interpretability:
> >
> > We agree that the term "interpretable" is quite loaded in the literature, and indeed, the interpretability research community points to the many different application-specific motivations and formulations of interpretability as a challenge in using the concept. (This point goes back at least to classic surveys like Doshi-Velez and Kim's "Towards a rigorous science of interpretable machine learning" and Lipton's "The mythos of model interpretability: In machine learning, the concept of interpretability is both important and slippery".) As such, there have been many calls to broaden the scope of what we might view as "interpretable" behavior, so as to expand the range of applications where it might be applicable. This has led, over time, to expansions of the scope in which interpretability is used -- for example, for purposes as diverse as searching for pathways to improve a model, finding flawed assumptions in a model, or providing people being evaluated by a model a mechanism for recourse in the face of the model's decision.
> >
> > It's in this spirit that we are implicitly proposing to add to the scope, without trying to supplant any of these valuable earlier formulations. In particular, the field of interpretability has done relatively less to conceptualize what it means for an action (as opposed to a prediction) to be interpretable, but this is the crucial question for the types of domains that motivate our work here. Additionally, complex settings such as our motivating domain of chess contain actions where there might be no succinct "explanation" (we know this both in practice, and also theoretically from the fact that general game-tree evaluation is believed to be outside the complexity class NP, and hence it is believed that there can be moves in a game that have no simple explanation).
> >
> > So if we want to extend notions of interpretability to the current setting, it cannot be through succinct explanations. This is why we propose that a natural extension of interpretable actions to a case such as ours is to ask whether a weaker agent is able to effectively follow up on the action of a stronger agent, to it, thus motivating our formulation of interpretability in a functional, as opposed to a semantic, notion. This is more acute in the HB framework, where the brains’ actions are to simplify the action-space (as opposed to taking the game to a more favorable state as could be said about STT), and we see it reasonable to use the word interpretability if the reduced action space leads to maia producing a better move. On an auxiliary note, we see that HB focals tend to agree with maia more than leela does, and that matches the notion of interpretability in that not only is maia likely to select better moves with the new prior enforced by our focals, but also moves that "agree" more with the focals in a sense.

---

> > > ### Comment · Reviewer_XZq8 · 2023-11-22
> > >
> > > Thank you for the thorough replies. I'll address a few points of particular interest in terms of the text structure. The answers here addressed by primary points of concern.
> > >
> > > > We agree that the term "interpretable" is quite loaded in the literature, and indeed, the interpretability research community points to the many different application-specific motivations and formulations of interpretability as a challenge in using the concept.
> > >
> > > I find it odd to frame atomic actions as interpretable by measuring how well they can be followed to achieve a shared, known goal (here: winning in chess). To me the definition is cleaner as "alignment" - showing that actions are more consistently aligned between collaborating agents toward a shared goal.
> > >
> > > That said, the motivation for this framing is much clearer from your answer. It might be good to slip some of this more explicitly into the text (and perhaps give the full response in the appendix or an expanded version of the text published elsewhere).
> > >
> > >
> > > > In response to the question concerning the inter-temporal nature of the definition of compatibility:
> > >
> > > Mentioning the car driving (for STT) and stock picking (for HB) examples would be valuable in the main text. Even if only a sentence or two to help contextualize the contribution.
> > >
> > >
> > > > We agree with you that testing att and tree is valuable on these agents, and we also understand the computational cost of the task. Accordingly, we add a previously omitted experiment where we run the other agents on these two players.
> > >
> > > Table 8 helps the case here greatly. The contrasting results with stockfish (Table 9) also point clearly toward the next open problems in this setting. Great! (If only these tables were in the main text...)
> > >
> > >
> > > > We have modified Section 4.2 to state the main takeaways more explicitly, and abstract the analysis to the appendix.
> > >
> > > This was a great improvement. If you need room to address other comments I feel this section can be mostly omitted without any great loss. The analysis is interesting, but I struggle to draw clear conclusions from it. The lessons are highly specific to this task and very nuanced, so it does not worry me to see them slip into the appendix. By contrast, the new section 4.4 is strong and should certainly remain.

---

> > > > ### Author Response · Authors · 2023-11-23
> > > >
> > > > Thank you for taking the time to go through our response and for your comments on the revised manuscript! AI alignment is definitely an interesting term that could 'align' well with the direction of our paper!
> > > >
> > > > We agree that the analysis is complex, and the takeaways from that particular section are very setting-specific. We chiefly perform them to ground our results, and get a sense of why we are obtaining the gains that we do. A plus is that they show there are multiple viable ways to achieve these gains (that is, the methodologies are not the same in effect).
> > > >
> > > > When making further revisions we will incorporate these additions; we do believe this discussion makes the scope and application of the work clearer

---

### Official Review · Reviewer_D1Jx · 2023-10-31

**Soundness:** 3 good
**Presentation:** 3 good
**Contribution:** 4 excellent
**Rating:** 8
**Confidence:** 4

**Summary:**

The paper considers the interaction between highly skilled agents with less-skilled counterparts. The paper uses chess as a benchmark due to the availability of both a variety of highly and less-skilled agents at a multitude of skill levels. The paper introduces three methodologies to compare a tree agent, an expector agent, and an attuned agent (finetuned self-play RL agent). Prior art has considered simultaneous teamwork between AI and humans at tasks but does not consider if they are inter-temporally compatible. The paper evaluates against state-of-the-art chess AI and finds that their combined method can outperform. The paper evaluates in two scenarios: random swapping between the high-skill and low-skill AI after each move and a variant of chess called “hand and brain”, which requires one teammate to select the piece to move while the other teammate chooses the move selection.

**Strengths:**

+The wide variety of chess bots available at different skill levels and variety of play styles at each skill level makes for a very interesting and realistic benchmark. I could see this having direct influence on human-AI teaming.

+The motivation for this paper is very strong. It is very interesting to consider the advice mismatch between superhuman AIs and mere human players. This is an open problem.

+Introduces a new version of interpretability: “interpretable iff a weaker agent can follow–up”

+Clear discussion of limitations.

**Weaknesses:**

-Although there are many comparisons between the lower-skilled agents and humans, this does not guarantee that the results will be the same with humans. It would be interesting to have a small experiment to confirm the results in this setting.

-The description of the main results do not claim that there is a best guidance for what another researcher should try as the main takeaway

**Questions:**

1) Tables 1-3 could be explained more clearly. Please consider bolding best results

2) Why is the EXP method N/A in table two?

3) Do you have any intuition as to why EXP performs best overall? Why does the paper not claim this as the best method if it has the highest winrate?

4) The results are explained well after reading through the experiments section many times. However, I believe that this could be clearer. Please consider using a chart that summarizes the best combinations (no score, just to guide the intuition).

---

> ### Author Response · Authors · 2023-11-19
>
> We thank you for taking the time to review our paper. We agree that a small human trial would be of interest in extending our concept of skill-compatibility to human-compatibility. And alongside further investigation of the concepts of measuring how skill-compatibility holds up under different styles and skill gaps, determining how skill-compatibility translates to actual human-compatibility is an open question worthy of further investigation. If a researcher wished to test our central result exemplified in table 1, we have uploaded the necessary weights and code onto our github for that purpose.
>
> In response to the specific questions:
> Tables 1-3 could be explained more clearly. Please consider bolding best results.
>
> Thank you for the suggestion. We have bolded the best results in Tables 1 and 2 (now merged for clarity) and rewrote the explanation for Table 3 for clarity.
>
> Why is the EXP method N/A in table two?
>
> The exp agent designed for HB only selects a piece based on its anticipation of maia’s actual move, and therefore is unable to actually play chess. The other agents are adapted to HB by simply masking the second part of the move, meaning Nf6 simply gets output as N in HB, with Nf6 still technically being there. exp simply outputs N, we don’t have anyway to get Nf6.
>
> Do you have any intuition as to why EXP performs best overall? Why does the paper not claim this as the best method if it has the highest winrate?
>
> Our intuitive guess as to why EXP performs well, especially in STT, is that accurately modeling the game for a few turns is more valuable than a long-term approximation in a highly noisy environment where the junior’s blunders are likely to throw the game off into different paths. The MCTS employed by other seniors refines the search assuming the opponent and the self are using a strong policy, but the presence of the weaker juniors means this behavior may not be not entirely accurate.
>
> The results are explained well after reading through the experiments section many times.
>
> In accordance with your suggestion, we have modified our results section to more clearly state our results immediately after the analysis.

---

> > ### Comment · Reviewer_D1Jx · 2023-11-22
> >
> > Thank you for the further clarifications. I think that the human experiments (even a small amount) would greatly increase the value of your work. Though, I agree that they are not a requirement for your work to be sufficient. Thank you for clarifying the writing. I will leave my score as is.

---

> > > ### Author Response · Authors · 2023-11-23
> > >
> > > Thank you for going over our clarifications! We agree that human experiments are a natural next step for a work of this nature, and we are pleased that you agree that the work is standalone even without human trials.

---

### Official Review · Reviewer_9rEp · 2023-11-01

**Soundness:** 3 good
**Presentation:** 3 good
**Contribution:** 3 good
**Rating:** 5
**Confidence:** 3

**Summary:**

The paper discusses the challenge of enabling powerful AI agents to interact effectively with agents of lower computational abilities. Using collaborative chess variants as models, the study proposes a framework to evaluate the compatibility of high-performing AI with lower-skilled entities. Traditional chess engines, though nearly optimal, struggled in this domain, leading to the development of three methodologies to create skill-compatible AI agents. These agents demonstrated superior performance in the proposed collaborative frameworks than conventional chess AI systems, such as AlphaZero. The paper emphasized that achieving raw performance in AI is not enough; the AI should also be compatible with lower-skilled agents for successful interactions.

**Strengths:**

- Innovative Concept: The paper introduces a novel and timely concept of "skill-compatibility" that addresses the real-world challenge of AI and human collaboration.

- Empirical Evidence: The use of collaborative chess variants as model systems offers practical insights and empirical proof-of-concept.

- Multiple Methodologies: The paper presents three distinct methodologies, showcasing the versatility of approaches to achieving skill-compatibility.
- Comparative Analysis: By comparing newly proposed agents with state-of-the-art chess AI, the research demonstrates the tangible benefits of their approach.

**Weaknesses:**

The main weakness is that the methods proposed don’t show a clear path toward broader human-AI collaboration. Since you use the weaker chess engine as a subroutine in search, it isn’t clear how you could make a version with humans since they can’t communicate at test time. Furthermore, these methods seem limited to chess. It would be more interesting to have methods that would work for a variety of cooperative or cooperative/competitive games such as Bridge or Hanabi.
Minor: formatting is off

**Questions:**

1. How can the methodologies be adapted for more dynamic, less rule-bound environments outside of chess?
2. Would the proposed techniques be effective in real-time scenarios where the lower-skilled entity is a human?
3. How does the proposed skill-compatible AI adapt to the continuous learning and evolving capabilities of lower-skilled agents?
4. In real-world applications, how can the balance between raw AI performance and skill-compatibility be optimized without compromising on crucial tasks?
5. Could the techniques be integrated into current AI frameworks for instant benefits, or would they require a complete overhaul of the system?

---

> ### Author Response · Authors · 2023-11-19
>
> Thank you for taking the time to review our paper. Below we address the points you brought up in your review.
>
> "Since you use the weaker chess engine as a subroutine in search, it isn’t clear how you could make a version with humans since they can’t communicate at test time."
>
> We apologize for the insufficient clarity on this point, which we have addressed in our updated manuscript. The attuned agents do not use the juniors as subroutines; they only train with them prior to testing. Thus one contribution of ours toward broader skill- and human-compatible collaboration is developing an agent that doesn’t need to communicate at test time. In response to your feedback and to that of the other reviewers, we have emphasized Section 4.4 and the points made within it to clarify the fact that our models do not need exact models of their opponents at either training or testing times. The updated section explicitly addresses your point concerning the ability of our agents to work without having an exact model of their juniors. Our results in this section show that even in an environment where we cannot communicate with the juniors and don’t have a correct model of them, generic approximations still perform well (e.g. one experiment takes expector seniors that use generic maia models as subroutines,  and they are able to beat leela when they matched with specific player models that they do not have access to).
>
> “These methods seem limited to chess. It would be more interesting to have methods that would work for a variety of cooperative or cooperative/competitive games”
>
> We start by noting that our methods could extend to other two-player zero-sum game games of perfect information, but we completely agree that developing methods for a broader spectrum of games, including cooperative games, is a worthy goal. Our belief is that doing so is a larger research agenda than one paper, so our approach here is to make tangible progress starting with chess. Like DeepMind did by starting with AlphaGo and then generalizing from there, we hope to start with the agents presented in this paper for chess and generalize to a broader set of domains.
>
> Questions
> How can the methodologies be adapted for more dynamic, less rule-bound environments outside of chess?
>
> This is a key question for the line of work that we hope this paper helps kickstart. Similar to how AlphaZero was a separate follow-up to the already-impressive AlphaGo work, adapting our methodology for chess to more dynamic, less rule-bound environments is its own worthy goal.
>
> Would the proposed techniques be effective in real-time scenarios where the lower-skilled entity is a human?
>
> We believe the answer is yes. Our closest approximation of this is playing versus specific human models partnered with generic seniors. In this setup, our agents do not have access to exact models of their junior partners (only to generic approximations). Although these player-specific models are not identical to individual human players, they are high-fidelity models of them, and they perform well. Real-time constraints are not an issue here, as moves are generated in the order of seconds by our agents.
>
> How does the proposed skill-compatible AI adapt to the continuous learning and evolving capabilities of lower-skilled agents?
>
> In this work we considered static agents, but as shown in section 4.4 the resulting models generalize to different agents (models of individual human chess players). As such, we expect that our methods could be adapted to work with continuously changing agents via periodic updates to the model.
>
> In real-world applications, how can the balance between raw AI performance and skill-compatibility be optimized without compromising on crucial tasks?
>
> This is a question that depends significantly on the domain. In chess AI performance is significantly greater than human so we are happy to trade a large amount of performance for skill-compatibility. If these methods were deployed in a real-world situation both the senior and junior agents would be required to act, e.g., the senior has an unpredictable delay between uses (an overtaxed server). In a situation like this obtaining compatibility would be of utmost importance as the junior agent might at any time have to complete the task with no support.
>
> Could the techniques be integrated into current AI frameworks for instant benefits, or would they require a complete overhaul of the system?
>
> In short, yes this can be integrated. Our methods require a good (but not perfect) model of humans to use for search/training. These models have become more and more common as training foundation models/semi-supervised training is very popular. If this model is available then our training loop can be easily integrated into an existing RL search/training framework. In fact, our own code was integrated into a Tensorflow MCTS search system without too much effort.

---

> > ### Comment · Reviewer_9rEp · 2023-11-21
> > **Reviewer Comment**
> >
> > Thanks for clarifying that you don't require a model of the opponent at test time. I've bumped my score but still vote to reject because I am not sure if there are many insights that generalize beyond this setting.

---

> > > ### Author Response · Authors · 2023-11-23
> > >
> > > Thank you for taking the time to go over and consider our clarifications. We do hope to see generalization to actual humans as well as different settings in future works!

---

### Official Review · Reviewer_SKaW · 2023-11-01

**Soundness:** 3 good
**Presentation:** 3 good
**Contribution:** 2 fair
**Rating:** 6
**Confidence:** 4

**Summary:**

This paper studies the problem of making super-human AI chess programs more "skill compatible" with humans, in that the moves they recommend are more likely to be moves that a human could successfully follow up and thus would actually have success playing in a game. This is studied by building on open-source engines like Leela.  The proposed methods either do not require training or only require fine-tuning versions of a powerful base model.

**Strengths:**

This is clearly an important problem and is, to my knowledge, quite understudied. This appears to be a natural set of approaches that should be tried, and the thorough empirical evaluation of these approaches is useful for the community.

They also appear to be the first to have formalized this problem, and they have set up about as good a methodology for evaluating their method as one could hope for short of human trials.

**Weaknesses:**

Given that these approaches do not train from scratch, there is some concern that the local improvements may not be representative of the improvements you would hope for if the models were retrained.

Section 4.4 should be emphasized more from the beginning. It seems the biggest challenge with this line of work is the model of their partner.  It's not enough just to match the "skill level" of the human; but it is also important to match the style of the human and adapt to the patterns in human mistakes. In the rest of the paper, it appears this aspect is ignored, but in Section 4.4, evaluation is done by fine-tuning to mimic specific human players and thus could hope to account for the exact sorts of errors those humans are more prone to. I believe this section greatly strengthens the work, and it should be emphasized to ensure the reader understands where much of the remaining difficulty is.

Minor points:
The “expector” agent appears to be quite similar to prior work[1,2], though without fine-tuning against the fixed opponent. It is also closely related to the opponent-exploitation work in the AI-poker community related to work like [3]. I would expect there to be some more similar work to this to be in the exploitability literature in game theory and in the adversarial attacks literature, investigating slightly different questions than this work.

[1] Timbers, F., Bard, N., Lockhart, E., Lanctot, M., Schmid, M., Burch, N., Schrittwieser, J., Hubert, T., and Bowling, M. Approximate exploitability: Learning a best response in large games.
[2] Wang, Tony Tong, et al. "Adversarial Policies Beat Superhuman Go AIs." (2023).
[3] Ganzfried, Sam, and Tuomas Sandholm. "Safe opponent exploitation."

**Questions:**

What is the relationship between this work and that of the exploitability and adversarial attacks fields?

---

> ### Author Response · Authors · 2023-11-19
>
> Thank you for your time and effort in reviewing our work, and for your support. Below we address the points you brought up in your review.
>
> “Section 4.4 should be emphasized more from the beginning”
> We agree that Section 4.4 was underemphasized in our original submission, and we will promote and expand it in our revision. We have uploaded a modified version of our manuscript demonstrating how we intend to clarify and expand it with analyses and text that we previously relegated to the appendix, as well as a small additional experiment. Thank you for this suggestion, we think this greatly helps the paper.
>
> Relation to opponent-exploitation work in AI poker:
>
> We agree that our work bears some similarities to the citations you listed, especially with regards to the use of a shallow MCTS to approximate the opponent agent at a particular moment of the game. One important difference is that the more sophisticated versions of adversarial MCTS deployed in these papers to pinpoint weaknesses in strong opponents are not necessary in ours to glean a noticeably advantage against the mixed opponent team (and in fact we notice that even 1-ply horizon EXP agents are still able to obtain good scores in the Appendix.) We will include the citations you listed and clarify this point in the main text.
>
> What is the relationship between this work and that of the exploitability and adversarial attacks fields?
>
> Thank you for pointing out that our work is related to the exploitability and adversarial attacks fields; we have updated the manuscript with citations to this literature. One main difference between this work and ours is that in our problem formulation, beyond adversarial play, we also have a junior partner agent on our side whom we wish to influence positively (or at the very least, protect from our adversaries), and we are accordingly not in full control of even our own team’s outputs. Another difference is in scope; these works develop technical methods to identify and exploit suboptimal play in sophisticated agents, whereas our agents sometimes learn simple ways to exploit a relatively simple agent (Maia).
> We find that it could be conceptually interesting to formulate the opponent team in a similar fashion to that done in [3], with the opponent being a mixed strategy that samples from a supposedly “optimal” strategy (that of the senior) and occasionally “gifts” when the junior plays.

---

> > ### Comment · Reviewer_SKaW · 2023-11-22
> > **Reviewer Comment**
> >
> > Thank you for your detailed response and for clarifying the relationship to related works. I'm glad you agree the more prominent treatment of Section 4.4 improves the paper. However, due to my remaining concerns and concerns raised by other reviewers, I will leave my score as is.

---

> > > ### Author Response · Authors · 2023-11-23
> > >
> > > Thank you for taking the time to go over our response and our revised manuscript, as well as for your literature suggestions!

---

### Author Response · Authors · 2023-11-19
**General Response**

We thank the reviewers for taking the time to provide thorough feedback and insightful reviews. We wish to use this space to address the chief item discussed in most reviews: the generalization experiments that test whether agents designed to work with general maia bots will be able to perform well against models that are fine-tuned to imitate players. After reading the reviews, we concur that it is perhaps an understated, yet critical point to the understanding of the paper. Accordingly, have reorganized it to both increase its scope and the clarity of the results within in accordance to some of your suggestions. The section will explicitly mention 3 types of cross-compatibility: cross-skill compatibility (1100 vs 1900 tests, moved from the appendix to the main paper), the specific players section expanded with a small extra test that had been omitted from the first paper, and finally, cross-style compatibility (using weakened stockfish, moved from the appendix to the main paper. To make room, we cut short the analysis for STT in the main paper, using this as an opportunity to clarify the main takeaways away from the details of the analysis which have been relegated to the appendix. We have already uploaded our revised manuscript (it is nearly identical in aspects beyond section 4, with the major changes being a shortening of 4.2 and an expansion of 4.4)

---

### Meta-Review · Area_Chair_QNW6 · 2023-12-05

**Metareview:**

This is an interesting paper studying how strong AI agents can be used to teach others. The reviewers recommended accepting this paper and I support their decision.

**Justification For Why Not Higher Score:**

The paper is very specific to chess and the results are somewhat limited.

**Justification For Why Not Lower Score:**

While specific, the results are interesting enough to recommend acceptance. People interested in chess and using AI for teaching will find this paper interesting.

---

### Decision · Program_Chairs · 2024-01-16

Accept (poster)